# Soil Carbon Stock and Indices in Sandy Soil Affected by Eucalyptus Harvest Residue Management in the South of Brazil

Jackson Freitas Brilhante de São José [1], Luciano Kayser Vargas [1,*], Bruno Britto Lisboa [1], Frederico Costa Beber Vieira [2], Josiléia Acordi Zanatta [3], Elias Frank Araujo [4] and Cimelio Bayer [5]

[1] Department of Agricultural Research and Diagnosis, Secretariat of Agriculture, Livestock, Sustainable Production and Irrigation of Rio Grande do Sul. 570, Porto Alegre 90130-060, Brazil; jackson-jose@agricultura.rs.gov.br (J.F.B.d.S.J.); bruno-lisboa@agricultura.rs.gov.br (B.B.L.)

[2] Soil and Forest Ecology Laboratory, Universidade Federal Do Pampa, São Gabriel 97307-020, Brazil; fredericovieira@unipampa.edu.br

[3] Embrapa Floretas, Estrada da Ribeira, Colombo 83411-000, Brazil; josileia.zanatta@embrapa.br

[4] CMPC, Celulose Riograndense, Rua São Geraldo, Guaíba 92703-470, Brazil; elias.araujo@cmpcrs.com.br

[5] Department of Soil Science, Faculty of Agronomy, Universidade Federal do Rio Grande do Sul, Porto Alegre 91540-000, Brazil; cimelio.bayer@ufrgs.br

* Correspondence: luciano-kayser@agricultura.rs.gov.br

**Abstract:** There has been limited research on the effect of eucalyptus harvest residue management on soil organic carbon (SOC) in subtropical environments. This research evaluated the effect on soil C indices of the following eucalyptus harvest residue managements: AR, with all forest remnants left on the soil; NB, where bark was removed; NBr, in which branches were removed; NR, which removed all residues; and NRs, which is same as NR but also used a shade net to prevent the litter from the new plantation from reaching the soil surface. C stocks within the soil depths of 0–20 cm and 0–100 cm increased linearly with the C input from eucalyptus harvest residues. In the layer of 0–20 cm, the lowest soil C retention rate was 0.23 Mg ha$^{-1}$ year$^{-1}$, in the NR treatment, while in the AR treatment, the retention rate was 0.68 Mg ha$^{-1}$ year$^{-1}$. In the 0–100 cm layer, the highest C retention rate was obtained in the AR (1.47 Mg ha$^{-1}$ year$^{-1}$). The residues showed a high humification coefficient ($k_1 = 0.23$) and a high soil organic matter decomposition rate ($k_2 = 0.10$). The carbon management index showed a close relationship with the C input and tree diameter at breast height.

**Keywords:** carbon management index; forest residues; soil health; plantation forestry; carbon sequestration

## 1. Introduction

Forest ecosystems have a high potential for soil carbon sequestration, around 0.4 Pg C year$^{-1}$ [1]. Among these ecosystems, eucalyptus plantations occupy roughly 25 Mha worldwide [2], of which 7.53 Mha are in Brazil [3]. In the next decade, this forest base should be expanded given Brazil's low carbon emission agriculture plan, which foresees an expansion of ~4 Mha by 2030 with planted forests, with the potential to mitigate 510 million Mg $CO_2$ eq of greenhouse gas emissions [4]. Much of this potential results from long-term carbon sequestration through increased soil organic matter (SOM) [5], which ensures the sustainability of forest production under subtropical conditions [6], especially in fragile soils such as sandy ones.

Forest companies have considered the removal of eucalyptus harvesting residues, given their possible use as an energy source [7]. Nevertheless, it is a practice that reduces nutrient availability [8,9], soil microbiota activity [10], soil quality [11], and forest productivity [11–14]. The impact of this practice on soil carbon stocks is still inconclusive. While certain research has indicated that the removal of eucalyptus residues can reduce soil carbon stocks [14–17], others have indicated that the impact is almost zero [18,19]. These

differences may be related to climate [20], the number of rotations in which the residue is removed from the area [14], soil texture and mineralogy [10,21], and the quality of the residue contributing to the soil [22,23].

Traditionally, it has been believed that recalcitrant residues were more efficient in accumulating C in soil, and recent evidence has revealed that higher-quality residues (lower C/N and lignin/N ratio) can show higher efficiency in accumulating C in soil than low-quality residues (higher C/N and lignin/N ratio) [24]. However, this effect depends on the available mineral surface for carbon stabilization, so that in sandy soils, much of the OM will be stabilized in the form of particulate organic C (POC) originating from more recalcitrant plant inputs [25]. Most of the eucalyptus harvest residues consist of branches and bark and are considered more recalcitrant because of the high content of phenolic compounds, notably tannin, lignin, and polyphenols [26]. These components are important in soil C retention in eucalyptus areas [10,17], especially in sandy soils characterized by a limited capacity of stabilizing C in soil organic matter [27].

Despite the substantial volume of information from long-term experiments concerning the effect of eucalyptus harvest residue management on the stocks of soil organic carbon [15,18,19], few initiatives have sought to take advantage of these data to estimate, for management, the rates of humification and soil organic matter losses. In this context, using a monocompartmental model based on first-order kinetics allows the prediction of the forthcoming development of SOC stocks until they stabilize and the amount of carbon that should be added to maintain the initial stocks [28].

In short or medium periods, the total organic C content is not a sensitive indicator to assess the effect of agricultural or forestry managements in the soil. Therefore, researchers have sought to combine the lability of organic matter in this evaluation by estimating the carbon management index (CMI). This index aggregates a carbon stock index (CSI) and a carbon lability index (LI), making it a sensitive tool to evaluate the impact of management practices, which also have a strong relationship with biological, chemical, and physical soil attributes [29,30]. The CMI has been efficient in evaluating crops and forest systems [31–33].

Our initial hypothesis is that the maintenance of eucalyptus harvest residues increases the retention of atmospheric C in soil organic matter after the rotation cycle compared to their removal. Hence, the objectives of this research were to assess the impact of managing eucalyptus harvest residues on the soil organic C content and stock, to evaluate eucalypt harvest residue management using the carbon management index and to determine the relationship between CMI and eucalyptus growth in sandy soil in subtropical Brazil.

## 2. Materials and Methods

### 2.1. Field Experiment and Treatments

The experiment was carried out in Rio Grande do Sul, the southernmost state of Brazil, in the city of Barra do Ribeiro. The experimental area was located around the coordinates 30°23′ S and 51°07′ W, with an altitude of about 30 m above the sea level. The climate in the local area is defined as humid subtropical (Cfa) according to the Köppen classification, and the region has an annual precipitation average of approximately 1400 mm, with no dry season. The highest average monthly temperature never exceeds 25 °C, and the lowest average monthly temperature is about 14 °C with slight frosts. The soil is a Quartzipsament, with sandy texture, weak structure, low capacity for water storage [34], and low cation exchange capacity (Table S1). The experiment was implemented in 2010 using *Eucalyptus saligna* (clone 2864). Each plot measuring 30 × 30 m was planted with 100 trees arranged in a grid of 10 rows and 10 plants per row. For the measurements, we considered an inner subplot of 18 × 18 m, containing 6 rows × 6 plants each. The experimental setup followed a complete randomized block design with four replicates and five treatments. The treatments were five eucalyptus residue managements, described as follows:

1.  AR: all forest residues remained on the soil (i.e., bark, branches, leaves, and the litter layer from the previous rotation), and only trunk wood was removed.
2.  NB: where bark was also removed.

3.   NBr: in which branches were also removed.
4.   NR: in which all eucalyptus residues were removed.
5.   NRs: same as NR, but a shade net was also used to prevent litter from the new plantation from reaching the soil surface.

*2.2. Estimation of Carbon Input and Biochemical Composition Analysis of Eucalyptus Harvesting Residue Components*

When the experiment was implemented, the branches, bark, and leaves from the previous rotation were collected and quantified in terms of their mass and ground. Based on information about accumulated litter production in experiments conducted by Celulose Riograndense with *Eucalyptus saligna* near the study region, the addition of litter accumulated until the sixth year of the current cultivation, was estimated using Equation (1), as adjusted by Witschoreck [35]:

$$\text{ABP (Mg ha}^{-1}) = 8.875950 + 0.100160 \text{ x (DBH x age)}, \tag{1}$$

where ABP is the accumulated litter production (Mg ha$^{-1}$), DBH is the average diameter at 1.30 m height obtained from trees, and age is the number of years since the eucalyptus plantation started. For C input by the eucalyptus roots, we considered the average data obtained by Londero et al. [36] at six years in *E. saligna* plantations in the nearby city of Guaíba, and an average C content of 37.84% from the data of Ribeiro et al. [37].

All components of the eucalyptus harvest residues were submitted to biochemical characterization. In the case of the litter from the current rotation, samples were collected from the NR treatment plots by placing a screen approximately one meter above the soil surface, and the litter was gathered about every two months. The C and N concentrations were determined by dry combustion in a Flash 2000 analyzer (Fisher Scientific Inc., Waltham, Massachusetts, EUA). Additionally, the lignin, cellulose, and hemicellulose contents of each component were determined following the conventional procedure described by Van Soest [38], in which the constituents of the plant tissue are separated via sequential filtration after heating with neutral and acid detergents. Hemicellulose is solubilized after the second filtration and is obtained by the difference between the two filtrates. The cellulose is calcined in a muffle, leaving the lignin.

*2.3. Soil Organic C Stock and Fractionation*

In July 2016, during the sixth year of the eucalyptus plantation, a 1 m deep and 1 m wide trench was opened per plot, and soil was collected on two opposite sides of each trench, totalizing eight samples per treatment in the following layers: 0–2.5, 2.5–5, 5–10, 10–30, 20–30, 30–50, 50–75, and 75–100 cm. As the spacing between plants was 3 × 3 m, and the trenches were opened at 0.5 m from a eucalyptus plant, one side of the trench configured the collection in the planting line, and the opposite side configured the collection between the planting lines. Equal amounts of soil from each sub-sample were pooled to form a composite sample representing the overall average of the spaces between rows and the planting row. The soil samples were oven-dried at 60 °C, grinded in a ball mill, and samples of ~50 g were removed. These subsamples were grinded again in a porcelain mortar and submitted to C analysis in a Shimadzu TOC-VCSH analyzer.

The bulk density in the same soil layer was evaluated by collecting metal rings [39]. Metal rings of 93.48 cm$^3$ were used in the 0–2.5 cm and 2.5–5 cm layers and 102.73 cm$^3$ for the other layers. The method of equivalent soil masses to the soil mass of the AR treatment was used to calculate soil organic C stocks [40]. Since the soil organic C contents of the experiment period in 2010 had been obtained using the wet digestion method (Walkley-Black), it was necessary to convert these contents to equivalent contents using the dry combustion method, which was the method used for the organic C analyses in this study. Thus, a set of 10 samples of this soil with a wide variation in organic C content was analyzed using the two methods, and a conversion coefficient of 0.66 was determined, which corresponds to the angular coefficient of the relationship between the C contents

determined via the wet combustion method and the total C contents via the dry combustion method (Figure S1).

The soil samples from the layer of 0–20 cm were also physically fractionated. For that, 20 g of air-dried and sieved to <2 mm soil and 70 mL of sodium hexametaphosphate solution were arranged in 100-mL capacity vials and were shaken in horizontal position for 15 h (60 cycles min$^{-1}$). The supernatant was sieved on a 53-µm sieve, and the fraction <53 µm, consisting of mineral-associated organic C (MAC), was dried at 50 °C, quantified in terms of its mass, and the organic C content was analyzed in a Shimadzu TOC VCSH analyzer. Particulate organic C (POC) was estimated as the difference between total organic C (TOC) and MAC (<53 µm) [41].

### 2.4. Carbon Management Index

The CMI was calculated in accordance with Blair et al. [29], assuming that the POC was labile C in the soil and the MAC was the non-labile fraction [42]. From this, the CMI is obtained by multiplying the C stock index (CSI) and the C lability index (LI) as follows:

$$\text{CMI} = \text{CSI} \times \text{LI} \times 100, \tag{2}$$

where CMI is the product of the CSI and LI. The CSI is obtained by the quotient between the stock of C in the treatment under evaluation and the stock of C in the soil of the reference treatment in the layer of 0–20 cm (Equation (3)). The LI (Equation (4)) is the quotient between the C lability (L) (Equation (5)) in the treatment under evaluation and the L in the reference treatment. NRs was considered to be the reference treatment for the estimation of CMI (CMI = 100).

$$\text{CSI} = \text{C stock in treatment}/\text{C stock in reference treatment} \tag{3}$$

$$\text{LI} = \text{L in treatment}/\text{L in reference treatment} \tag{4}$$

$$\text{L} = \text{POC}/\text{CAM} \tag{5}$$

### 2.5. Annual Soil C Retention Rates

Annual soil C retention rates were calculated by subtracting the C stock in the treatment from C stock in the NRs treatment (Equation (6)), considering a period of six years.

$$\Delta \text{ C soil (Mg ha}^{-1}\text{ yr}^{-1}) = (\text{C soil treatment} - \text{C soil NRs})/(6 \text{ years}) \tag{6}$$

### 2.6. Estimate of the Organic Matter Humification ($k_1$) and Decomposition ($k_2$) Coefficients

The fraction of added C retained in soil organic matter ($k_1$), called the humification coefficient, was roughly calculated from the angular coefficient of the linear regression relating the amounts of C added annually to the annual rate of variation (dC/dt) in the soil organic C stock in the 0–20 cm layer.

From the values of the effective addition of C to the soil by the management of eucalyptus harvest residues ($k_1$A) and the stocks of organic C in the 0–20 cm soil layer, we estimated the annual rate of soil organic matter loss ($k_2$) using the equation dC/dt = $k_1$A-$k_2$C in the condition dedC/dt = zero according to the process reported by Bayer et al. [43] and Vieira et al. [30]. In this condition, $k_1$A = $k_2$C and $k_2$ = $k_1$A/C, where C represents the soil organic C stock in the initial condition (13.02 Mg ha$^{-1}$) and A represents the annual C addition rate required to maintain the initial soil organic C stock unchanged over time (i.e., dC/dt = zero).

### 2.7. Statistical Analysis

The effects of the treatments on soil organic C levels were evaluated using analysis of variance and the difference between the means of the treatments using the Tukey test at 5%.

Linear regression analyses were utilized to verify the relationship between C addition in the different managements of eucalyptus harvest residues and the stocks of total organic C and mineral-associated organic C (POC and MAC), as well as concerning CSI, LI, and CMI. Furthermore, linear regression analyses were employed to gain deeper insights into the relationship between CMI and tree diameter at breast height (DBH). On the soil sampling date, we assessed 36 trees from each plot to estimate these parameters.

### 3. Results

#### 3.1. C Contribution and Composition of Eucalyptus Harvest Residues, Litter, and Roots

The C contribution to the soil varied according to the eucalyptus harvest residue management; it varied from 2.04 Mg ha$^{-1}$ year$^{-1}$ in the NRs treatment, in which all residues were removed, to 5.04 Mg ha$^{-1}$ year$^{-1}$ in the AR treatment, in which all residues but the trunk wood were kept (Table 1). Of these C contributions, leaves totaled 0.03 Mg ha$^{-1}$ year$^{-1}$, branches accounted for 0.96 Mg ha$^{-1}$ year$^{-1}$, bark 0.45 Mg ha$^{-1}$ year$^{-1}$, the litter from the current rotation totaled 1.56 Mg ha$^{-1}$ year$^{-1}$, and roots totaled 2.04 Mg ha$^{-1}$ year$^{-1}$.

**Table 1.** Total biomass and C contribution to soil and biochemical composition of harvest residues from the previous eucalyptus rotation and litter from the current eucalyptus rotation.

| Eucalyptus Residue Management | Litter [a] | Composition of Harvest Residues | | | | |
| --- | --- | --- | --- | --- | --- | --- |
| | | Leaves | Branches | Bark | Root | Total |
| | | Residue biomass (Mg ha$^{-1}$ year$^{-1}$) | | | | |
| NRs | 0.00 | 0.00 | 0.00 | 0.00 | 5.41 [b] | 5.41 |
| NR | 3.05 | 0.00 | 0.00 | 0.00 | 5.41 | 8.46 |
| NBr | 3.05 | 0.07 | 0.00 | 1.10 | 5.41 | 9.62 |
| NB | 3.05 | 0.07 | 2.17 | 0.00 | 5.41 | 10.70 |
| AR | 3.05 | 0.07 | 2.17 | 1.10 | 5.41 | 11.80 |
| | | Carbon input (Mg ha$^{-1}$ year$^{-1}$) | | | | |
| NRs | 0.00 | 0.00 | 0.00 | 0.00 | 2.04 [c] | 2.04 |
| NR | 1.56 | 0.00 | 0.00 | 0.00 | 2.04 | 3.60 |
| NBr | 1.56 | 0.03 | 0.00 | 0.45 | 2.04 | 4.08 |
| NB | 1.56 | 0.03 | 0.96 | 0.00 | 2.04 | 4.59 |
| AR | 1.56 | 0.03 | 0.96 | 0.45 | 2.04 | 5.04 |
| | | Biochemical composition | | | | |
| C (g kg$^{-1}$) | 501.07 ± 25.97 | 474.77 ± 7.1 | 442.53 ± 2.7 | 413.35 ± 9.2 | - | - |
| N (g kg$^{-1}$) | 11 ± 0.1 | 15.74 ± 0.7 | 1.99 ± 1.1 | 3.74 ± 0.2 | 2.8 [d] | - |
| C/N | 47.1 ± 2.0 | 30.2 ± 0.9 | 316.2 ± 255.8 | 110.6 ± 9.0 | 162.7 [e] | - |
| Lignin (g kg$^{-1}$) | 274.4 ± 0.3 | 227.1 ± 2.8 | 142.5 ± 2.7 | 87.1 ± 0.3 | - | - |
| Lignin/N | 25.8 ± 2.0 | 14.4 ± 1.9 | 112.3 ± 111.0 | 23.2 ± 0.8 | 82.9 [e] | - |
| Hemicellulose (g kg$^{-1}$) | 104.4 ± 0.3 | 122.2 ± 0.2 | 192.4 ± 0.4 | 158.6 ± 1.0 | - | - |
| Cellulose (g kg$^{-1}$) | 204.4 ± 0.9 | 143.3 ± 0.9 | 580.2 ± 2.4 | 450.1 ± 1.5 | - | - |

[a] Addition estimated according to Witschoreck [35]; [b] root addition according to Londero et al. [36]. [c] C content in roots 37.84%, obtained from Ribeiro et al. [37], [d] obtained by Guimarães et al. [44], and [e] obtained by Demolinari et al. [17].

The values of C/N and lignin/N ratio in the eucalyptus harvest residue management are shown in Table 2. In general, we observed that the C/N ratio values in the eucalyptus residue managements ranged from 111.7 in the NBr treatment to 162.7 in the NRs treatment. As for the lignin/N values, the values ranged from 58.1 in the NR management to 69.1 in the NB treatment.

#### 3.2. Effect of Eucalyptus Harvest Residue Management on Soil Organic C Content and Stocks

Eucalyptus harvest residue managements had no statistically significant effect on soil organic C levels (Figure S2). Nevertheless, there was a trend for C contents to increase with the maintenance of the eucalyptus harvest residues. When the layers up to 10 cm depth

were averaged, the soil in the NRs treatment presented an organic C content that was 39% lower than that observed in the AR treatment.

The soil organic C stocks of the 0–20 cm ($r^2 = 0.80$, $p = 0.03$) and 0–100 cm ($r^2 = 0.82$, $p = 0.03$) layers increased linearly with the increase in C input by the maintenance of eucalyptus harvest residues (Figure 1a,b). Each 1 Mg ha$^{-1}$ year$^{-1}$ of residue resulted in a differential accumulation of 1.41 and 2.95 Mg ha$^{-1}$ in the layers of 0–20 cm and 0–100 cm, respectively, at the end of 6 years of cultivation. The partial and total maintenance of the eucalyptus harvest residues in the soil (NBr, NB, and AR) determined higher stocks of C than in the other management methods, presenting, in the layer from 0 to 20 cm depth, organic C contents that were 30, 31, and 27% higher than in the treatment in which the harvest residues from the current harvest were completely removed (NR) (Figure 1a). These findings demonstrate that these three types of management, despite having low quality components, such as bark and branches, resulted in higher soil organic carbon stocks.

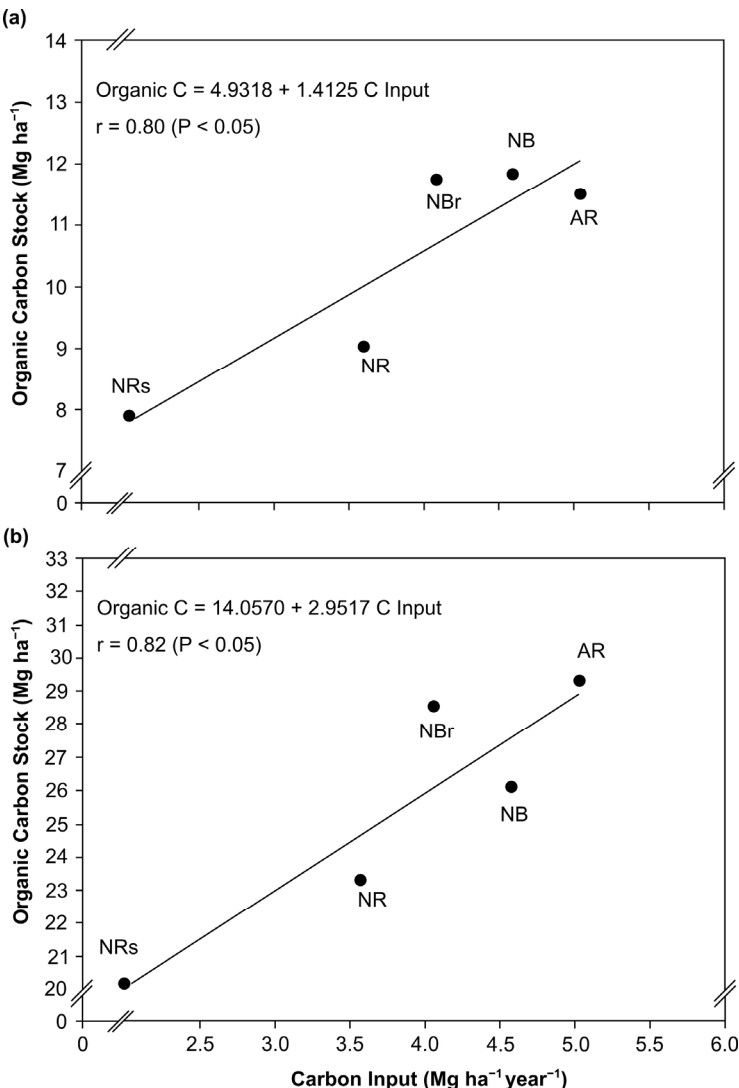

**Figure 1.** Relationship between C input by eucalyptus harvesting residue management and organic C stock in the 0–20 cm depth layer (**a**) and in the 0–100 cm depth layer (**b**).

**Table 2.** C/N and lignin/N ratio of eucalyptus harvesting residue management.

| Residue Management | C/N [1] | Lignin/N [1] |
|---|---|---|
| NRs | 162.7 | 82.9 |
| NR | 112.6 | 58.1 |
| NBr | 111.7 | 53.9 |
| NB | 154.6 | 69.1 |
| AR | 150.7 | 65.0 |

[1] Ratio weighted by the amount of waste added in each treatment.

### 3.3. Annual Soil C Retention Rates

Annual soil C retention rates varied from 0.23 to 0.68 Mg ha$^{-1}$ year$^{-1}$ in the 0–20 cm layer (Figure 2a). In this soil layer, the NR treatment exhibited the lowest soil C retention rate (0.23 Mg ha$^{-1}$ year$^{-1}$), while with the AR treatment, the retention rate was 0.68 Mg ha$^{-1}$ year$^{-1}$ (i.e., about threefold greater). In turn, the maintenance of the bark (NBr) and branches (NB) determined accumulation rates that were quite similar to treatment AR. Similar trends were observed in the 0–100 cm layer, with the annual soil C retention rates reaching 1.47 Mg ha$^{-1}$ in the AR treatment. In the intermediate treatments with the maintenance of bark (NBr) or branches (NB), annual soil C retention rates were 0.8 and 1.0 Mg ha$^{-1}$ year$^{-1}$, respectively (Figure 2b). Although few studies have related soil C retention rates to forest residue management, maintaining residues on the soil surface is one of the main approaches to increase soil C retention rates in various rotations. The annual soil C retention rate went from 0.5 to 1.6 Mg ha$^{-1}$ in the 0–100 cm layer, reinforcing that more than half of the C stabilization occurs in the 20–100 cm layer (Figure 2b).

### 3.4. Estimate of the Humification ($k_1$) and Decomposition ($k_2$)

Figure 3 shows the relationship between the amount of C inputted (A) by the management of eucalyptus harvest residues and the annual rate of change (dC/dt) of the organic C stocks in the 0 to 20 cm layer in relation to the initial stock of organic C at the beginning of the field experiment (13.02 Mg ha$^{-1}$). The angular coefficient of the equations represents the $k_1$, that is, the fraction (or percentage) of added C that effectively remains in the soil. The value of $k_1$ is 0.23, meaning that approximately 23% of the added C was integrated into soil organic matter after one year. Knowing the value of $k_1$, the decomposition rate ($k_2$) was estimated to be 0.10 year$^{-1}$ (i.e., ~10% of the soil C is released to the atmosphere as $CO_2$ by microbial decomposition).

### 3.5. Carbon Lability, Carbon Management Index, and Relationship with Eucalyptus Growth

The highest C stock index (CSI) values were obtained with full (AR) and partial (NBr and NB) maintenance of eucalyptus harvest residues, which promoted 46, 50, and 49% increases, respectively, regarding the reference (NRs) treatment. On the other hand, the NR treatment increased CSI by only 14% compared to NRs management (Table 3 and Figure 4). There was a linear relationship between POC and the addition of C by eucalyptus harvest residue management. The mineral-associated organic C (MAC) showed reduced values compared to POC and with a slight tendency to increase values with C input by the management of eucalyptus harvest residues. The higher proportions of organic C in POC form in the soil in NBr and AR determined higher lability (L) and, consequently, LI compared to NRs (Table 3). The LI in the 0–20 cm depth layer was associated with the C input from the eucalyptus harvest residue management and ranged from 1.00 to 1.35 (Table 3). Higher values of CMI resulted in a higher diameter at breast height (DBH) of eucalyptus, as shown by the linear regression between the two variables ($r^2 = 0.98$, $p < 0.001$) (Figure 5).

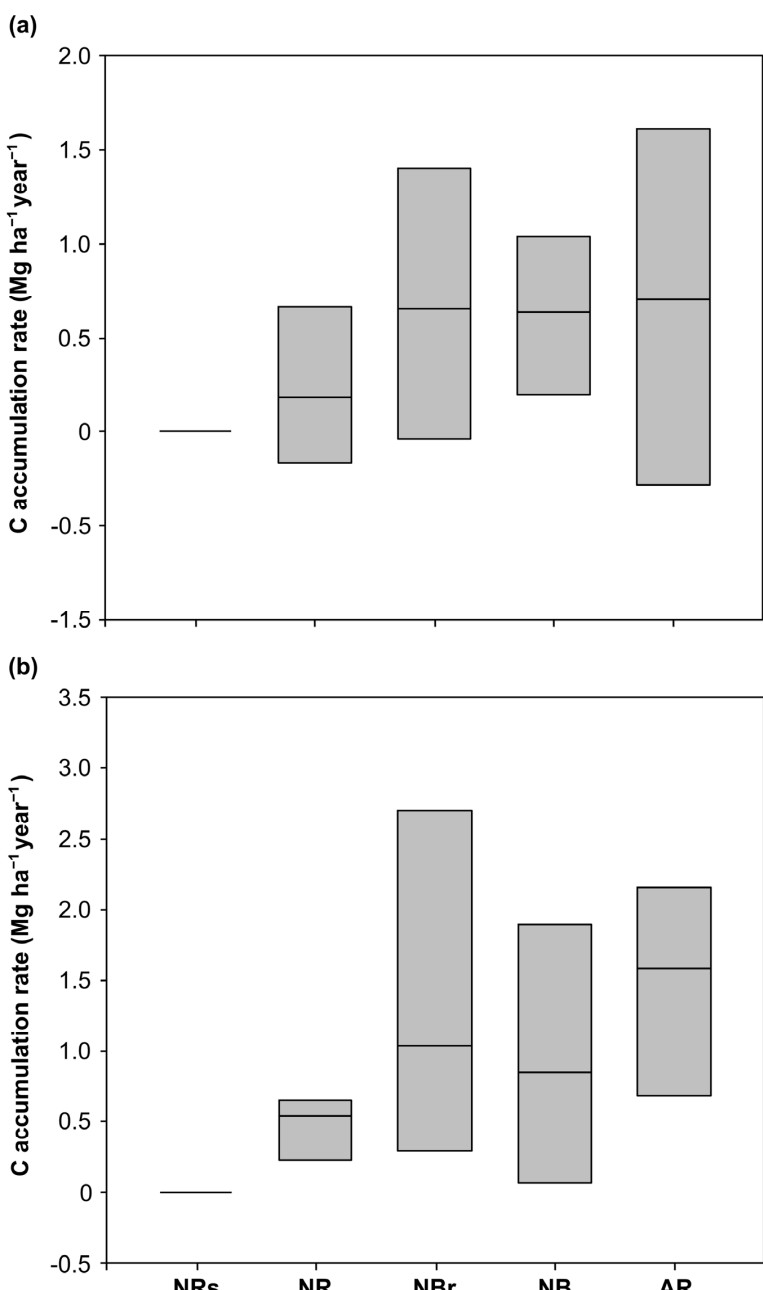

**Figure 2.** Annual soil C accumulation rates in the 0 to 20 cm (**a**) and 0 to 100 cm (**b**) layers cultivated under different eucalyptus harvest residue managements at six years of age. Boxplot shows minimum, average, and maximum values.

**Table 3.** Soil particulate organic carbon (POC), mineral-associated organic C (MAC), C lability (L), C lability index (LI), C stock index (CSI), and C management index (CMI) in the 0 to 20 cm layer of an Quartzipsament cultivated under different eucalyptus harvest residue management, at six years of age.

| Management | POC | MAC | L | LI | CSI | CMI |
|---|---|---|---|---|---|---|
| | ——— g kg$^{-1}$ ——— | | | | | |
| NRs | 2.92 | 0.033 | 88.4 | 1.00 | 1.00 | 100 |
| NR | 3.33 | 0.032 | 104.0 | 1.17 | 1.14 | 133 |
| NBr | 4.32 | 0.036 | 120.0 | 1.35 | 1.50 | 202 |
| NB | 4.47 | 0.038 | 117.6 | 1.33 | 1.49 | 203 |
| AR | 4.42 | 0.039 | 113.3 | 1.28 | 1.46 | 186 |

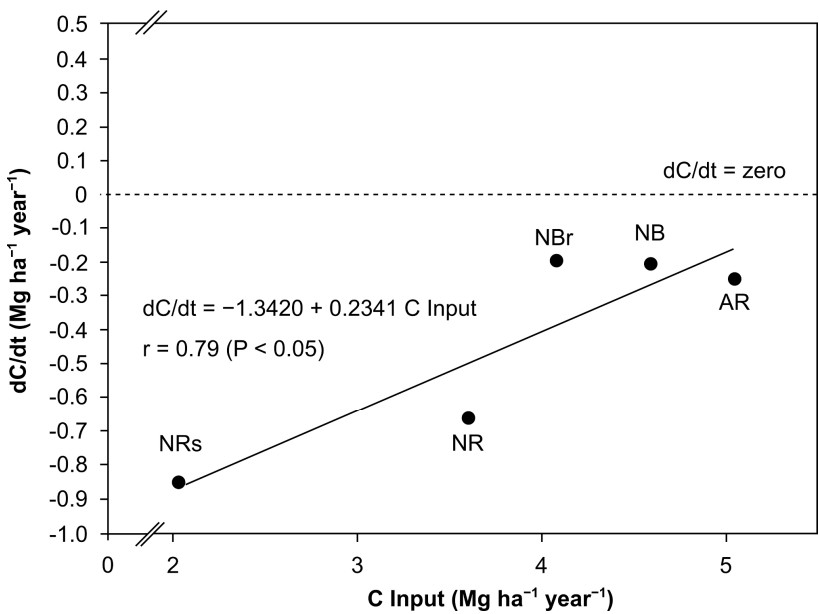

**Figure 3.** Relationship between the amounts of C added and the variation (dC/dt) of the soil organic C stocks in the 0–20 cm layer cultivated under different managements of eucalyptus harvesting residues at six years of age.

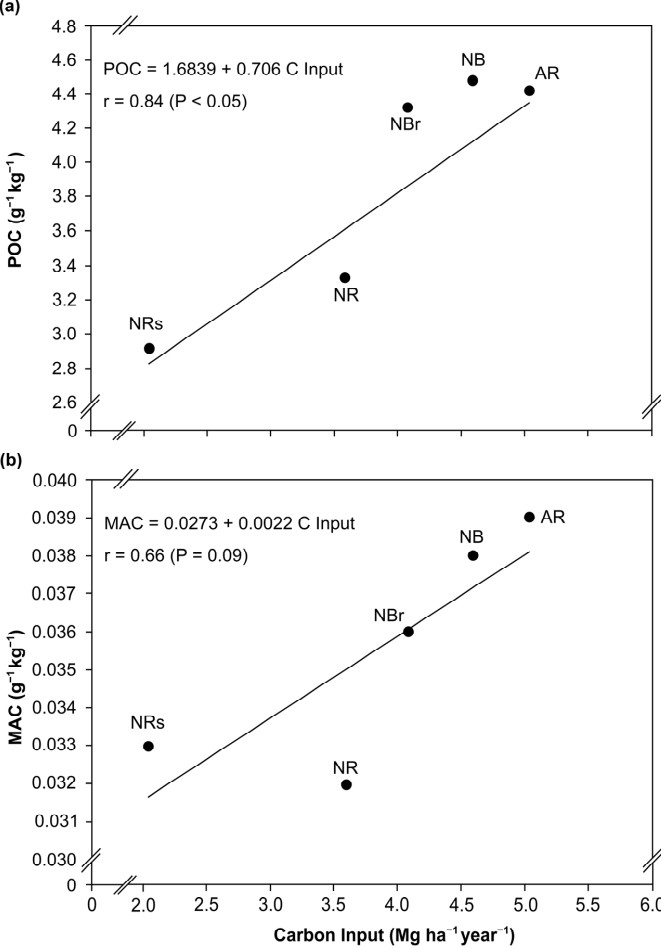

**Figure 4.** Relationship between the amounts of carbon added and the particulate organic C (POC) (**a**) and the mineral-associated organic C (MAC) (**b**) of the SOC stocks in the layer of 0–20 cm of a soil cultivated under different managements of eucalyptus harvesting residues at six years of age.

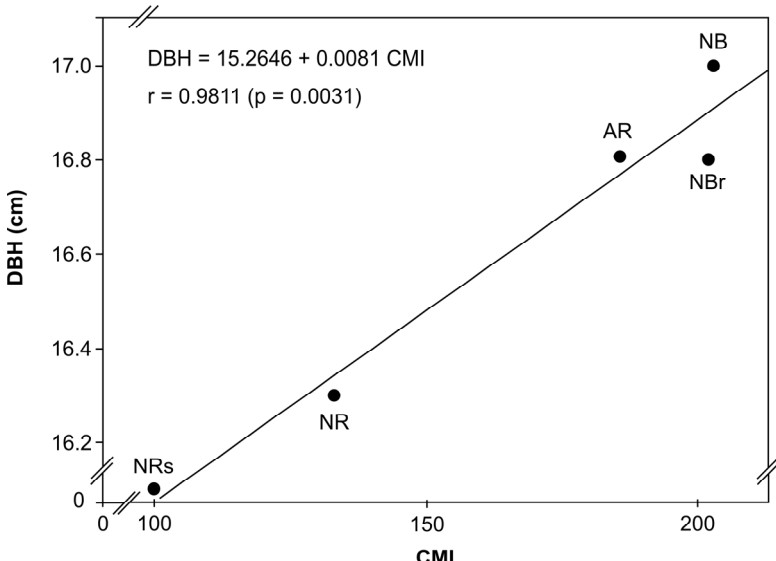

**Figure 5.** Relationship between the C management index of the soil (CMI) and the diameter at breast height (DBH) under different managements of eucalyptus harvesting residues at six years of age.

## 4. Discussion

The eucalyptus harvest residue managements resulted in different quantities of C inputted annually into the soil. As hypothesized, the input of C was inversely proportional to the removal of harvest residues, resulting in the AR exceeding the NRs by 3.0 Mg ha$^{-1}$ year$^{-1}$. These results closely resembled the ones obtained by Demolinari et al. [17]. The authors emphasized the significance of retaining the eucalyptus harvest residues in the field for the maintenance of soil C, particularly the bark, as it constitutes the most abundant component of harvest residues.

Our results also corroborate a study conducted on sandy soils under eucalyptus in Congo in which eucalyptus harvest residue removal also altered soil C contents by ±40% [15]. However, in soils with elevated clay content, this effect could not be observed in the first production cycle [19,44], requiring more than one cycle with successive removal for this effect to be observed [18]. This occurs due to the higher efficiency of the physical C protection mechanism in clayey soils, which reduces the access of the microbial community and, thus, the mineralization rate of SOM [28,45]. Compared with the NR treatment, maintaining the bark (NBr) and branches (NB) promoted increments of 18 and 44% in organic C contents in the 0–2.5 cm soil layer, respectively. Nonetheless, comparisons with the NRs and NR treatments revealed that the removal of the forest litter promoted a 49% reduction in the SOC contents in the 0–2.5 cm soil layer. These results possibly have to do with the amount and the composition of the C contributed by the different components of the eucalyptus harvest residues.

The litter is a constituent that presents in its chemical composition a low C/N ratio in comparison to the bark and branches, although it has higher lignin levels (Table 1). The impacts of the quantity and composition of the C contributed by forest residues on soil C retention can be variable. On the one hand, the removal of eucalyptus litter promotes a reduction in soil C contents [46]. On the other hand, a study by Wang et al. [23] demonstrated that, irrespective of the amount and the composition of the litter contributed by eucalyptus species on the soil surface, there is no impact on the SOC contents. This is because the contribution in quantity and quality of C in soils under forests may not always consistently lead to an increase in SOC due to differences in the biochemical composition of residues, in soil mineralogy, and in physical protection mechanisms of SOC [47].

Typically, litter removal in forested areas reduces the organic C contents in surface and subsurface soil layers [48]. This practice causes more significant changes in organic C content in tropical and subtropical soils [49]. In fact, this was verified by Sena et al. [50],

who observed, in sandy soil, a decline in the carbon content and stock in the 20 to 40 cm layers after 90 days of complete or partial eucalyptus litter removal, evincing that removing the litter that would accumulate during six years could rapidly reduce the soil carbon stock. Likewise, our results demonstrate that the top layers of sandy soils are extremely sensitive to removing residues and litter. This practice may directly affect soil C storage in this subtropical ecosystem.

Our results indicate that roughly half of the contributed C was accumulated in the 20–100 cm layer. This may be linked to the contribution of recalcitrant residues and the presence of the deep root system of eucalyptus [1,51]. Roots are more efficient than leaf litter in increasing organic carbon stocks in the soil profile. This effect was demonstrated by Pegoraro et al. [52], who observed that carbon contribution from eucalypt roots equals 50% of the total harvest residue. Furthermore, dissolved organic C can alleviate the soil profile and stabilize subsurface layers more efficiently than surface soil layers due to the higher soil saturation deficit [53]. However, in this extremely sandy soil (33 g kg$^{-1}$ clay), it is unlikely that the organic–mineral interaction is an effective mechanism for stabilizing organic C in soil, with organic matter stabilization being more related to its biochemical recalcitrance [27].

In the treatments NBr, NB, and AR, the maintenance of the eucalyptus harvest residues resulted in higher soil organic carbon stocks, despite the low quality components, such as bark and branches. Therefore, in the first moment, these results lead us to believe that there is a divergence from the theory that higher quality residues, characterized by fast mineralization rates, narrow C/N ratio, and reduced phenol concentrations, lead to greater microbial efficiency in carbon accumulation in the soil compared to low-quality residues, characterized by low mineralization rates and higher C/N and elevated phenol concentrations [24].

When analyzing the annual soil C retention rates, we observed that more than half of the C stabilization occurred in the 20–100 cm layer. This result reinforces the need for sampling of deep soil layers (up to 100 cm), since sampling restricted to superficial layers (20 cm) may underestimate the environmental impact of forest management systems. Similar findings have been obtained in agricultural areas, where some researchers have demonstrated that C accumulation in the 20–100 cm soil layer is equivalent to 30–50% of the C accumulated in the 0–30 cm layer [32]. However, studies related to soil C retention in areas with *Eucalyptus* have concentrated on the superficial soil layers and have no thickness standardization, making it challenging to establish comparisons between different studies. In areas of *Eucalyptus*, Lima et al. [54] found an average annual retention rate of 0.22 Mg ha$^{-1}$ year$^{-1}$ of C over thirty years in the 0–10 cm layer, with the highest rate obtained between the second and third rotation in areas of altitude reaching 0.57 Mg ha$^{-1}$ year$^{-1}$ of C. In contrast, Cook et al. [55] and Hernández et al. [56] found a retention rate of 0.20 Mg ha$^{-1}$ year$^{-1}$ of C in the 0–15 cm layer in eucalyptus plantations. Possibly, the results of these studies are underestimated due to the sampling being restricted to the superficial soil layers. Thus, subsurface layer sampling is recommended for determining C retention rates in soils under *Eucalyptus* forests.

The obtained value of $k_1$ of 0.23 year$^{-1}$ is well above the average values of $k_1$ in agricultural soils [30,43]. The intrinsic characteristics of the residue, such as its origin (aerial part and root) and composition (lignin content and C/N ratio), will directly affect the soil's residence time. In this sense, possibly, the quality of the eucalyptus harvesting residues contributing to the soil in the present study favors a higher humification coefficient due to the lower lability of the material, given the presence of recalcitrant components, like bark and branch, with an elevated C/N ratio (Table 2). Maintaining these residues on the soil surface has resulted in a higher C transfer from these components to the soil [17]. In addition, studies have shown that roots have a humification coefficient 2.3 times higher than surface-contributed residues due to the higher C/N ratio [57].

Regarding the values of $k_2$, the value of 0.10 year$^{-1}$ was much higher than the average values of $k_2$ of 0.019 and 0.040 year$^{-1}$ obtained in agricultural soils under no-till and con-

ventional tillage, respectively [43]. The sandy soil texture is possibly one of the main factors related to this high annual decomposition rate, because of the diminished aggregation and physical protection of organic matter and the low stabilization capacity by organic–mineral interactions [45].

Full (AR) and partial (NBr and NB) maintenance of eucalyptus harvest residues resulted in the highest C stock index (CSI) values. These results reinforce the significant contribution of eucalyptus harvest residue maintenance to soil C accumulation [14,58]. Almost all the organic carbon in eucalyptus harvest residue management is particulate organic carbon (POC) (Table 3). This proportion of POC is higher than that obtained in a study carried out by Oliveira Filho et al. [59] in forest and sugarcane areas under Quartzarenic Neosol, who observed that approximately 60% of total organic C is in the form of POC. This difference between the results may be related to the fact that their study presented clay contents six times greater than in our study, which could favor a higher stabilization of C in the mineral fraction of the soil.

The carbon lability index (LI) expresses the ability of management systems to preserve labile SOM compared to the reference management [60], and it was related to the C input from the eucalyptus harvest residue management in our research. This effect was also evidenced by Rocha et al. [14] in eucalypt areas, who observed that removing eucalypt harvest residues for two consecutive rotations reduced the soil's labile C fraction. Thus, these results demonstrate the significance of maintaining the eucalyptus harvest residues to preserve the SOM.

The LI of NBr and AR were 32% higher than NRs. These results show that although the bark and branch components present in the NBr and AR managements are of low quality, they favored the LI in the soil. The increase in LI values observed in the current investigation may be related to biochemical recalcitrance, which contributes to carbon accumulation in the labile fraction [61], since the soil in the present study has a low mineral surface area available for soil C stabilization [62]. These effects were also found by Puttaso et al. [63], who observed a more significant increase in soil carbon in the labile fraction when the frequent application of residues with moderate contents of nitrogen, lignin, and polyphenols and low cellulose contents was performed in sandy soil. This fraction has been utilized as an indicator to evaluate alterations in the SOM arising from the change in use or management practices in forest systems [33].

The higher CSI and LI were reflected in the CMI. The CMI is an index that compares the alterations in CSI and LI, reflecting the C sequestration and nutrient cycling potential. LI was less sensitive than CSI for these two indices, and this can be seen in the smaller range of values for LI (35%) compared to CSI (50%). On average, the three managements with the highest CMI (NBr, NB, and AR) together had a CMI that was 97% higher compared to the reference management NRs, demonstrating that an improvement in the soil quality has occurred, and indicating that forest systems in which eucalyptus harvest residues are maintained on the soil surface are more sustainable in the long term. Therefore, this index is an excellent tool for assessing the soil quality and production systems [30].

The highly significant linear regression between the DBH and CMI demonstrates that the CMI is a reliable quality index for soil management evaluation and show that both lability and carbon stocks are important in soil sustainability and maintenance of eucalyptus growth. The CMI efficiently estimates the influence of conservation management practices [31]. Management practices that allow the return of residues to the soil surface contribute to increasing CMI values. In fact, this was observed in an investigation by Chatterjee et al. [64], who observed that maintaining 10 Mg ha$^{-1}$ of wheat residues in maize-grown areas resulted in a higher CMI when compared to sites without residues. An improvement in the CMI has resulted in increased yields of maize and rice [32]. Thus, maintaining eucalypt harvest residues is a forest management strategy that improves the SOM and forest productivity [11,14,46].

Given our results, we can classify the AR, NB, and NBr as superior for soil quality and NRs and NR as inferior. Thus, adopting management strategies that promote the removal

of bark and branches to favor silvicultural operations or the use of these components for bioenergy production is a forest management strategy that should be avoided by forest companies, especially in fragile soils such as sandy ones.

## 5. Conclusions

The maintenance of eucalyptus harvest residues is a strategy that promotes increases in soil carbon stocks in sandy soils over the removal of harvest residues. The bark and branches are components that help to increase the rates of carbon retention of the soil, and it is crucial to maintain these components in the field. Therefore, extremely sandy soils depend more on the contribution of recalcitrant residues (bark and branches) that provide great resistance to decomposition by chemical recalcitrance.

The soil humification coefficient under eucalyptus harvest residue management was high, as was the annual rate of SOM loss. The CMI is a sensitive indicator to evaluate the quality of eucalyptus harvest residue management, showing a close relationship with the addition of C and tree diameter at breast height, demonstrating that management that reduced the soil quality also reduced eucalyptus growth.

**Supplementary Materials:** The following supporting information can be downloaded at: https://www.mdpi.com/article/10.3390/soilsystems7040093/s1, Table S1. Soil chemical and physical attributes in the experimental area; Figure S1: Relationship between C evaluated by Walkley-Black (wet digestion) method and dry combustion method; Figure S2. Total organic carbon content of a Quartzarenic Neosol in the 0–100 cm layer, cultivated under different management of eucalyptus harvest residues at six years of age.

**Author Contributions:** Conceptualization, J.F.B.d.S.J., L.K.V., B.B.L., F.C.B.V., J.A.Z., E.F.A. and C.B.; methodology, J.F.B.d.S.J., E.F.A., and C.B.; formal analysis, J.F.B.d.S.J., F.C.B.V., J.A.Z. and C.B.; investigation, J.F.B.d.S.J., F.C.B.V., J.A.Z., E.F.A. and C.B.; resources, E.F.A.; data curation, J.F.B.d.S.J.; writing—original draft preparation, J.F.B.d.S.J. and L.K.V.; writing—review and editing, J.F.B.d.S.J., L.K.V. and B.B.L.; supervision, C.B.; project administration, J.F.B.d.S.J. and C.B.; funding acquisition, E.F.A. All authors have read and agreed to the published version of the manuscript.

**Funding:** This research was funded by Celulose Riograndense—CMPC, Foundation for Research Support of Rio Grande do Sul State (Fapergs, Innovation and Technology Network of Low Carbon Agriculture and adapted to Climate Change in Rio Grande do Sul State), the National Council for Scientific and Technological Development (CNPq), Research Centre for Greenhouse Gas Innovation (RCGI), hosted by the University of São Paulo (USP), and sponsored by FAPESP—São Paulo Research Foundation (2020/15230-5) and Shell Brasil.

**Institutional Review Board Statement:** Not applicable.

**Informed Consent Statement:** Not applicable.

**Data Availability Statement:** The data presented in this study are available on request from the corresponding author. The data are not publicly available due to being part of a broader study in progress.

**Acknowledgments:** The authors are grateful to CMPC for enabling assessment of the field experiment and for funding.

**Conflicts of Interest:** The authors declare no conflict of interest.

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
