# Peer review of "Soil Carbon Stock and Indices in Sandy Soil Affected by Eucalyptus Harvest Residue Management in the South of Brazil"

_soilsystems, doi:10.3390/soilsystems7040093_

Round 1

Reviewer 1 Report

1-      The study is clear and well-framed in terms of the questions addressed.

2-      It would help to add a few sentences about quality assurance in the analysis methods in lines 127-130.  Adding about 10 lines concerning the equipment used and the conduct of the procedures would provide assurance the analysis did correspond to standard methods, Don’t overdo it but the authors should ensure the accuracy of the results.

3-      Line 206 “eucalyptus harvest residues management” not clear

4-      Figure 1 is hard to read, perhaps a change in scale or the style of graph would help.  The results in Fig. 1 indicate non-significant differences where there are some clear trends, this suggests the sample sizes were likely too small considering the high variability and patchiness in most soils in terms of organic matter content. On the other hand, the r2  values for the regressions in  fig. 2 are more than adequate to demonstrate trends (which is what one would expect). Perhaps Fig. 1 should be dropped or reinterpreted.

5-      Line 365 – separated superscript – fix the spacing

6-      The sandy soils are very important to the outcome of the research, yet there is little discussion of them in the methods other than a sentence of general description.  It would be helpful to know more about their structure and history and to know how common these soils are and where they occur in the region of interest.  Are they commonly planted with eucalyptus? Are they frequently degraded by agriculture and then abandoned or left to fallow?  A full paragraph on the soils would help to provide a more robust context for the conclusion. 

Author Response

We thank the reviewer for the valuable contributions. We tryed to incorporate all of them in the text.
We complemented the description of the method as requested. We also dropped the Figure 1 from the main text. 

Reviewer 2 Report

The authors have conducted a very large and important study. As a recommendation for future research, I advise analyzing carbon sequestration not only by soil layers, but also by soil genetic horizons, which will more fully reflect ecosystem processes.

Author Response

We thank the reviewer for the valuable contributions. 

Reviewer 3 Report

This is a well-organized manuscript, providing new insight on soil carbon sequestration under eucalyptus, as affected by harvest residue management.

Minor revision is requested before it can be accepted for publication.

Below,  several comments are provided, which may be helpful during the revision.

Title.

Please, consider the “stock” in the title. For example: “ Soil carbon stock and indices in sandy soil affected by eucalyptus harvest residue management in southern Brazil”.

Abstract.

In general abstract should be ameliorated.

Please, see also Comments on the Quality of English Language

Introduction

Lines 62 to 68. The idea or significance of coefficients k1 and k2 should be more thoroughly addressed.

Lines 79 to 85. Please, consider to reduce the high number of particular objectives. May be some objectives could be merged.

Material and Methods.

Lines 89 to 90. Please, indicate the altitude of the experimental area.

Line 94. Please, provide a citation for Arenosol (IUSS, 2015).

Results.

Table 1 is difficult to understand, namely the biochemical composition.

Conclusions

This section should be enhanced and ameliorated. Please, take into account the objectives to rewrite conxlusions.

The English Language should be tightened in the different sections of the manuscript.

Next, there are several examples of mistakes. But this lis is not exhaustive. Please, edit all the text.

In general, there are several redundancies. For example, “addition C input” (line 24). Please, delete addition.

Line 19. Soil organic carbon is abbreviated by (C). However, in Line 40 soil organic matter is abbreviated by SOM. It would be more consistent SOC and SOM.

Sentence is line 19 to 23 is too long. Rewritten recommended.

Sentences in lines 25 to 27 are not clear.

Line 40. Please, delete levels. Seems redundant.

Etc., Etc.

….

Lines 223 and 224. “are shown” instead of “are showed”.

Etc., Etc.

Author Response

We thank the reviewer for the valuable contributions. We tryed to incorporate all of them in the text.
We complemented included the word 'stock' in the title.
We tryed to improve the Abstract.
We changed the phrase about k1 and k2 and the particular objectives.
 We tryed to improve the English quality.

Reviewer 4 Report

The paper is about an interesting and essential topic of carbon sequestration in the soil and meets the journal's scope. On the other hand, it has serious methodological flaws and is therefore unsuitable for publication. My main concerns are about the applied approach. Authors partly took values as standards for calculations published in national journals and not understandable for foreigners, or even as theses without any peer review. Moreover, they use a constant to estimate soil organic carbon content based on organic matter measurements. There are no details about the SD value related to this constant. This is too vague to accept. I think the applied approach is not scientific enough for a Q1 paper. In the further parts, additional estimations were done again based on publications that were not provided. Thus, I have stopped reading the paper at this point.

My further comments, notes, and questions are in the attached text.

Author Response

We thank the reviewer for the valuable contributions. We tryed to incorporate all of them in the text.
Regarding the equations we used, we agree that it would be better if they have been published in international journals. However, those equations were developed in the same site of our research. The authors of those equations belong to a traditional and respected research team based on Federal University of Santa Maria. We have no reasons to descredit their results.

Round 2

Reviewer 4 Report

The paper is still based on a national approach that is not available to an international reader.

Moreover, the authors did not provide an item-by-item response to my comments but gave a general statement.

Author Response

We thank the reviewer for the valuable contributions. We tried to incorporate all of them in the text.

Regarding the equations we used, we agree that it would be better if they have been published in international journals. However, those equations were developed in the same site of our research. The authors of those equations belong to a traditional and respected research team based on Federal University of Santa Maria. We have no reasons to discredit their results.

Here follows a  point-by-point response to your review:

Linha 56: And what about particulate organic matter? Light fraction may be a considerable part of SOC stock. Check e.g. https://www.nature.com/articles/s41467-023-38700-5

-We inserted a comment about POC and used the suggested reference: “However, this effect depends on the available mineral surface for carbon stabilization, so that in sandy soils much of the OM will be stabilized in the form of particulate organic C (POC) originating from more recalcitrant plant inputs [25]”

Angst, G.; Mueller, K.E.; Castellano, M.J.; Vogel, C.; Wiesmeier, M.; Mueller, C.W. Unlocking complex soil systems as carbon sinks: multi-pool management as the key. Nat. Commun. 2023, 14, 2967. https://doi.org/10.1038/s41467-023-38700-5

Linha 59: Please avoid the term "quality". You are talking about chemical composition.

Accepeted. We deleted that fragment.

Linha 65: please cite the introduction of these measures

-We changed the phrase (few initiatives have sought to take advantage of these data to estimate, for management, the rates of humification and soil organic matter losses. In this context, using a monocompartmental model based on first-order kinetics allows the prediction of the forthcoming development of SOC stocks until they stabilize and the amount of carbon that should be added to maintain the initial stocks [28].)

Linha 70: Soil forestsà misleading term. It can refer to the origin as a soil type, and also used for soils under the wood. Please specify.

-We changed to “In short or medium periods, the total organic C content is not a sensitive indicator to assess the effect of agricultural or forestry managements in the soil.”

Linha 70 à why? There are many studies against this statement

-The phrase refers to short or medium periods of time, in which TOC will not change significantly. So, in this case, it is not a sensitive indicator, and C fractions that are more sensitive to soil management may serve as early indicators of changes in soil C dynamics (see https://doi.org/10.1016/j.geoderma.2014.09.002).

Linha 78: What do you mean by retention? There is no clue in the introduction if POC is included or not.

-Retention means keeping, maintenance, the opposite of releasing. POC is included and it is mentioned now in introduction.

Linha 88: What was the age of the plantation at harvest? What was the plan before?

-Six years. It appears in line 122 of the first version of the manuscript.

Linha 95 à (supplementary material). à not provided

-We are sorry for that. We included it now.

Linha 113 - was it the same species?

-It is the same species. It was a bad translation, sorry for that. We changed the text for “previous rotation”.

Linha 118: This is a not peer-reviewed study in Portuguese; thus, an average reader can not check its validity.

-We agree. On the other hand, it is a model that was adjusted for local conditions and it is the model that the company uses for years. Again, we have no reasons to discredit it.

Linha 122: Again, it is a paper without an English abstract; thus, I can not check it. I am afraid if the axioms of the approach can not be validated, the whole work is undermined.

-The abstract is also in English. The paper was peer-reviewed. Anyone that really wants to read it can easily use a translator and have at least a good idea about the research.

Here is an option:

https://www-scielo-br.translate.goog/j/cflo/a/b8ZwnkbDPXp3NbVPmj5Nh4H/?lang=pt&_x_tr_sl=pt&_x_tr_tl=en&_x_tr_hl=pt-BR&_x_tr_pto=wapp

Linha 130 à  Van Soest [38].

We described the method.

Linha 141: These subsamples à Any replications?

The field replications, as previously described.

Linha 142: to organic C analysis by dry comà what about inorganic carbon content? No mention of that. This method measures total carbon.

We correct that. We changed to “total C”

Regarding inorganic C, this soil has no significant amounts of it, if have any. According to Naorem et al. (2022), four conditions are essential for the formation of inorganic C: “(a) high soil pH (alkaline), (b) an active source of CO2 in soil for HCO3− production, (c) a large amount of available Ca2+ and (d) an optimum level of soil moisture”.

In our research, soil pH varied from 4.9 to 5.1, and Ca availability is extremely low, 0.5 to 0.8 cmolc dm-3. So, there is no possibility to exist IC in this soil.

Naorem et al. (2022): https://doi.org/10.3390/agriculture12081256

Linha 145: metal rings àReplications?

Four replications, as previously stated.

Linha 151: This is too vague to accept. I think the applied approach is not scientific enough for a Q1 paper.

At least you would provide the results of the 10 measurements.

-We provided it now in the supplementary material.

Linha 161: Particulate organic C (POC) à The introduction completely neglected this term and the related field.

We fixed that, as suggested.

Linhas 216 a 218: These references are not provided. Therefore I believe the whole calculation is theoretical without any solid evidence linking to reality.

I can not find these without providing the numbers

The references were provided. We just forgot numbering the references in the footnote. We are sorry for that. You could have found them by using “Ctrl+F” and writing the word you wanted to search (Demolinari, for instance).

Linha 245: Are the symbols single values or averages? If so what were the SD? If not there is no reason to compare single values.

Quartzoarenic soil à It was classified as Arenosol in the methods.

We removed the figure from the manuscript, as suggested by other reviewer.

We standardized the soil classification, Thank you for calling our attention to that.
